

# A Python-enhanced urban land surface model SuPy (SUEWS in Python, v2019.2): development, deployment and demonstration

Ting Sun[1], Sue Grimmond[1]

[1]Department of Meteorology, University of Reading, Reading, RG6 6BB, UK

*Correspondence to*: Ting Sun (ting.sun@reading.ac.uk); Sue Grimmond (c.s.grimmond@reading.ac.uk)

**Abstract**. Accurate and agile modelling of the climate of cities is essential for urban climate services. The Surface Urban Energy and Water balance Scheme (SUEWS) is a state-of-the-art, widely used, urban land surface model (ULSM) which simulates urban-atmospheric interactions by quantifying the energy, water and mass fluxes. Using SUEWS as the computation

kernel, SuPy (SUEWS in Python), stands on the Python-based data stack to streamline the pre-processing, computation and post-processing that are involved in the common modelling-centred urban climate studies. This paper documents the development of SuPy, which includes the SUEWS interface modification, F2PY (Fortran to Python) configuration and Python frontend implementation. In addition, the deployment of SuPy via PyPI (Python Package Index) is introduced along with the automated workflow for cross-platform compilation. This makes SuPy available for all mainstream operating systems

(Windows, Linux, and macOS). Furthermore, three online tutorials in Jupyter notebooks are provided to users of different levels to become familiar with SuPy urban climate modelling. The SuPy package represents a significant enhancement that supports existing and new model applications, reproducibility, and enhanced functionality.

## 1  Introduction

Cities have the highest need for climate-resilient environments given their large and ever-increasing populations. One

prerequisite to building climate resilience is climate information at various spatio-temporal scales, for example to understand: energy partitioning over urban surfaces (Li et al., 2015b; Sun et al., 2017; Wang et al., 2015; Ward and Grimmond, 2017; Zhao et al., 2014), pedestrian level meteorology to diagnose thermal comfort (Bar et al., 2011; Erell et al., 2013; Krayenhoff et al., 2018; Sun et al., 2016; Tan et al., 2009), or ambient radiation and wind conditions to assist building design (Chen, 2004; Jentsch et al., 2013; Li et al., 2015a; Lindberg and Grimmond, 2011; Reinhart and Cerezo Davila, 2016; Santamouris et al.,

2001). To obtain such climate information, accurate and agile modelling capacity of the urban climate is essential.

Urban land surface models (ULSM) are widely used to simulate urban-atmospheric interactions by quantifying the energy, water and mass fluxes between the surface and urban atmosphere (Best and Grimmond, 2015; Chen et al., 2011; Wang et al.,



2012). These models require information on urban morphology (e.g., heights, spacings of buildings, etc) and anthropogenic dynamics (e.g., building-operation-related heat release, emissions of heat by traffic) to be included.

One widely used and tested ULSM, the Surface Urban Energy and Water balance Scheme (SUEWS) (Table 1), requires basic meteorological data and surface information to characterise essential urban features (i.e., urban surface heterogeneity and anthropogenic dynamics). SUEWS enables long-term urban climate simulations without specialised computing facilities (Järvi et al., 2011; 2014a; Ward et al., 2016). SUEWS is regularly enhanced (Grimmond and Oke, 1986; 1991; Grimmond et al., 1991; Offerle et al., 2003; Järvi et al., 2011; Loridan et al., 2011; Järvi et al., 2014; Ward et al., 2016; Järvi et al., 2019) and tested in cities under a range of climates worldwide (Table 1). Although operationally simple and scientifically robust, SUEWS still requires some skill for application (e.g., computing environment setup, parameter configuration, etc.), which may limit uptake for broader applications in urban planning and design.

**Table 1: Recent studies using SUEWS.**

| Topic | City | Reference |
|---|---|---|
| CO2 emission module development | Helsinki, Finland | Järvi et al. (2019) |
| Impacts of anthropogenic heat and irrigation on surface energy balance | Shanghai, China | Ao et al. (2018) |
| Sensitivity of SUEWS to forcing variables | Vancouver, London | Kokkonen et al. (2018) |
| Description of SUEWS as the core processor of the UMEP system | (N/A) | Lindberg et al. (2018) |
| Impacts of changes in surface cover, human behaviour and climate on energy partitioning | London, UK | Ward and Grimmond (2017) |
| Offline evaluation of SUEWS driven by WRF output | Porto, Portugal | Rafael et al. (2017) |
| Implications of warming to cold climate cities. | High latitudes cities | Järvi et al. (2017) |
| Evaluation in Singapore and comparison with other urban land surface models | Singapore | Demuzere et al. (2017) |
| Evaluation in four cities under different background climates | Dublin, Ireland; Hamburg, Germany; Melbourne, Australia; Phoenix, USA | Alexander et al. (2016) |
| Evaluation of SUEWS in two UK cities | London and Swindon, UK | Ward et al. (2016) |
| Evaluation of radiation flux in Shanghai | Shanghai, China | Ao et al. (2016) |
| Boundary layer modelling and coupling with SUEWS | Sacramento, USA | Onomura et al. (2015) |



| | | |
|---|---|---|
| Evaluation with Local Climate Zone information as surface characteristics | Dublin, Ireland | Alexander et al. (2015) |
| Model inter-comparison for sensible and latent heat fluxes | Helsinki, Finland | Karsisto et al. (2015) |
| Snow melt model development and evaluation | Helsinki, Finland; Montreal, Canada | Järvi et al. (2014b) |
| SUEWS development and initial evaluation | Vancouver, Canada; Los Angeles, USA | Järvi et al. (2011) |

Reproducibility and open science principles are more increasingly important (Peng, 2011). Although climate scientists by convention publish detailed model configurations used in their research, minor inconsistencies or lack of transparency of code often hampers efforts to reproduce simulation results. In addition, new users may lack prerequisite knowledge in low-level
compilation and scripting to undertake initial model runs and interpretation of simulation results (Lin, 2012).

Python is now extensively by the atmospheric sciences community for data analyses and numerical modelling (Lin, 2012; Perkel, 2015) thanks to its simplicity and the large scientific Python ecosystem (e.g., PyData community: https://pydata.org). Recent Python-based endeavours include global climate system models (Monteiro et al., 2018), stochastic geological models
(Varga et al., 2019), hydrological models (Hamman et al., 2018), to cite just a few.

In this paper, we present a Python-enhanced urban climate system based on the popular Fortran-coded SUEWS - SuPy (SUEWS in Python). The development of SuPy (Section 2), the essential workflow in its cross-platform deployment (Section 3), and three demonstrations tutorials for users of different levels (Section 4) are presented.

## 2   Development

The following are considered within the design process of SuPy:

1) *Data preparation*: Climate simulations typically require extensive pre-processing of data (loading input data, reformatting to conform with standards, etc.) and post-processing (conversion of output, graphical and cartographic plotting, etc.).
Python has a vast array of utilities to support this; notably, NumPy (the fundamental package for scientific computing with Python, https://www.numpy.org) and pandas (a tabular-format-centred data analysis tool) are two cornerstone libraries in Python-based scientific computing.

2) *Performance:* Python, as a scripting language, has poorer performance than compiled languages (e.g., C, Fortran) (Kouatchou, 2018). For this reason, Fortran is used extensively for weather and climate related software (e.g., WRF





(Skamarock and Klemp, 2008), GFDL AM3 (Donner et al., 2011), etc). Therefore, by using different languages their strengths can be utilised.

3) *Cross-platform ability*: Given the range of computer environments, it is important that software can be easily used across operating systems (OS) with ease. Python and Fortran both are easily used on most OS. However, for performance reasons as noted, it is preferable to have a compiled backend for intensive simulations, where platform specific compilations are mandatory. Thus, we adopt the Microsoft Azure Pipelines to allow cross-platform ability for SuPy (Section 3.1).

4) *Extendibility*: It is desirable, possibly even essential, for the scientific model to interact with other models and data sources to extend the overall capacity and to explore urban climate related questions beyond the climate science.

To address these four considerations, SuPy's architecture uses Python's data processing and Fortran's computational efficiency. SuPy consist of three parts (Figure 1):

1) *SUEWS*: a Fortran-coded local scale urban land surface model of moderate complexity that can simulate the urban surface energy balance in combination with the complete urban hydrological cycle, considering irrigation and runoff processes (Grimmond and Oke, 1986, Grimmond and Oke 1991, Grimmond et al. 1991, Offerle et al. 2003, Järvi et al. 2011, Jarvi et al. 2014, Ward et al. 2016).

2) *SuPy_driver*: calculation kernel compiled by F2PY (Fortran to Python, part of the NumPy package) (Peterson, 2009) to facilitate the transfer of SUEWS Fortran modelling ability to Python and guarantee the computational performance.

3) *SuPy:* a Python-based frontend processor using pandas `DataFrame` and derived functionality for data analysis and simulation management.

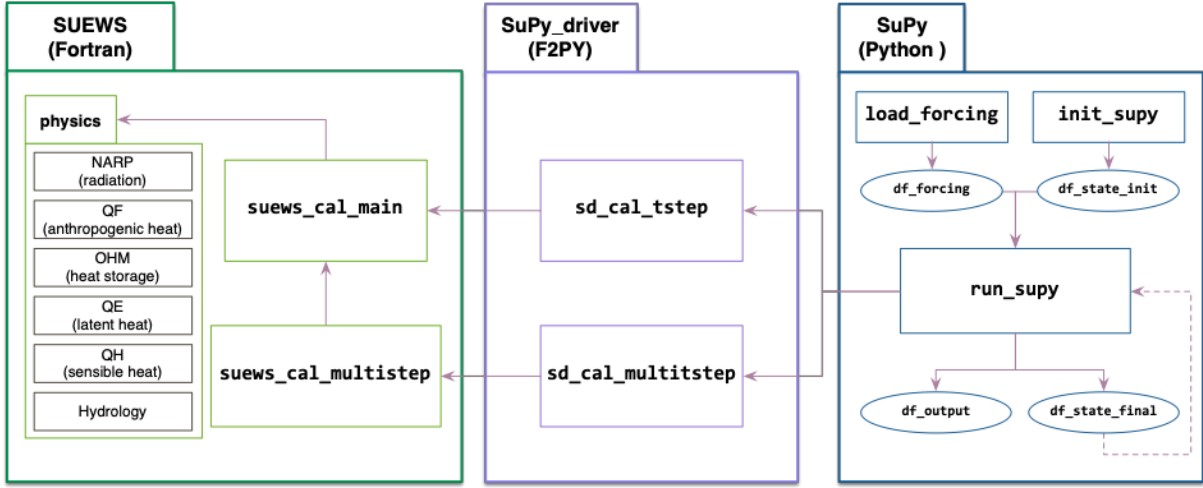

**Figure 1: SuPy's three major components (left to right): a) SUEWS, a Fortran-coded local scale urban land surface model; b) SuPy_driver, the calculation kernel compiled by F2PY; c) SuPy, a Python-based frontend processor.**



Development of SuPy started with SUEWS v2017b (Ward and Grimmond, 2017). As SUEWS has mostly been operated as a stand-alone land surface scheme, there have been few demands for the model code to interact with other models in a coupled environment. Thus, to develop SuPy significant reconfiguration of the model interfaces was required. Although the physical modules were already separated, the first step was to explicitly separate the model physics from the input/output (I/O) modules
of SUEWS.

Key structural changes involved reducing the use of Fortran modules to pass variables and parameters and return to more use of Fortran subroutine arguments with intent (e.g. in, out) explicitly stated. The modified physics subroutines are called from two subroutines `suews_cal_main` and `suews_cal_multistep` (Fig. 1a) depending on the model timestep (single or multi-).
This structure constitutes the SUEWS v2018b (Sun et al., 2018) calculation kernel (Figure 1a) and enables efficient communication between SUEWS and other models (e.g., WRF) through an explicit and unified interface.

Enhancements to SUEWS modelling capability in v2018b includes the surface diagnostics module to produce pedestrian level meteorological variables ($T_2$: air temperature at 2 m above ground level (agl), $q_2$: specific humidity at 2 m agl, and $U_{10}$: wind
speed at 10 agl). These variables are particularly useful for urban climate service applications given their widespread use for indicating climate state (Tan et al., 2009). From similarity theory (Monin and Obukhov, 1954) we estimate:

$$T_2 = T_s - \frac{Q_H}{k u_* \rho C_p}\left[\ln\left(\frac{z - d_0}{z_{0h}}\right) - \Psi_h\left(\frac{z - d_0}{L}\right) + \Psi_h\left(\frac{z_{0h}}{L}\right)\right] \tag{1}$$

$$q_2 = q_s - \frac{Q_E}{k u_* \rho L_v}\left[\ln\left(\frac{z - d_0}{z_{0v}}\right) - \Psi_v\left(\frac{z - d_0}{L}\right) + \Psi_h\left(\frac{z_{0v}}{L}\right)\right] \tag{2}$$

$$U_{10} = \frac{u_*}{k}\left[\ln\left(\frac{z - d_0}{z_0}\right) - \Psi_m\left(\frac{z - d_0}{L}\right) + \Psi_m\left(\frac{z_0}{L}\right)\right] \tag{3}$$

where $T_s$ ($q_s$) is surface temperature (humidity), $Q_H$ ($Q_E$) the sensible (latent) heat flux, $k$ the von Karman constant (0.4), $u_*$ the friction velocity, $\rho$ the air density, $C_p$ the air heat capacity, $L_v$ the latent heat of vapourization, $z$ the diagnostic height, $z_0$ ($z_{0h}/z_{0v}$) the surface roughness length for momentum (heat/vapour), $d_0$ the zero-plane displacement, and $\Psi_m$ ($\Psi_h/\Psi_v$) the
integral form of stability correction function for momentum (heat/vapour), the chosen forms are the same as in previous versions of SUEWS (Järvi et al., 2011).

The SUEWS v2018b kernel is compiled by F2PY to generate the Python-compliant `SuPy_driver` package. Using the two subroutines allows better computational performance. The SuPy_driver calls the two subroutines depending on timestep
simulation type: single (`sd_cal_tstep`) or multi-timestep (`sd_cal_multitstep`, Figure 1b). The former is useful in flexible manipulation of SuPy runtime behaviours (application in Section 4.3), while the latter has much better performance because of the much lower computational overheads with the F2PY wrapper. Therefore, `sd_cal_multitstep` is the





default executer in `run_supy`, the SuPy core processor performing simulations, for regular runs without runtime manipulation.

In addition to `run_supy`, SuPy uses pandas DataFrame as the central data structure to simplify the pre- and post-processing

as required by the original SUEWS. The pre-processor is designed to load existing SUEWS input files, which consists of:

1) `init_supy`: This loads surface characteristics (e.g., albedo, emissivity, land cover fractions, etc.; for full details refer to SUEWS documentation at: https://suews-docs.readthedocs.io/en/latest/input_files/SUEWS_SiteInfo/SUEWS_SiteInfo.html) and model configurations (e.g., stability correction function; for full details refer to SUEWS documentation at: https://suews-

docs.readthedocs.io/en/latest/input_files/RunControl/RunControl.html) into `df_state_init` (a pandas DataFrame, for full details refer to SuPy documentation at: https://supy.readthedocs.io/en/latest/data-structure/df_state.html). Two auxiliary `json` files are used with the look-up hierarchies for loading this information from SUEWS library files (details refer to SUEWS documentation at: https://suews-docs.readthedocs.io/en/latest/input_files/input_files.html) in a consistent file-code-variable way.

2) `load_forcing`: Meteorological and other external forcing information are loaded into `df_forcing` (a pandas DataFrame, for full details refer to SuPy documentation at: https://supy.readthedocs.io/en/latest/data-structure/df_forcing.html) to drive the SuPy simulations with timestep size inferred from its `DatetimeIndex` (i.e., the `freq` attribute). SUEWS should be run at short timesteps (e.g. 5 mins) as precipitation or irrigation runoff from impervious surfaces becomes too large if the water arrives as one large hourly (or longer) amount (Grimmond and Oke,

1991; Ward et al., 2018). As such, `load_forcing` is implemented with the ability to downscale the raw forcing data to finer timesteps (5 min by default).

Although detailed guidance is provided in SUEWS documentation for preparing input files (see https://suews-docs.readthedocs.io/en/latest/input_files/input_files.html), it is often very time intensive to get these files prepared

properly, in particular for new users. To ease the preparation of the input file, a helper function `load_SampleData` is provided in SuPy to get the sample input DataFrames (i.e., `df_state_init` and `df_forcing`) ready for running simulations. These DataFrames can also be used as templates to prepare input data for SuPy without wrestling with multiple input files, which can be useful for both existing and new users to SUEWS/SuPy. Examples for using the sample datasets are provided as tutorials (section 4).


As the F2PY-compiled kernel SuPy_driver relies on NumPy `ndarray` for data input and output, two SuPy post-processors `pack_state` and `pack_output` are embedded in `run_supy` to pack the `ndarray` output of model final states and simulation results to `df_state_final` (a pandas DataFrame, full details refer to SuPy documentation at:



https://supy.readthedocs.io/en/latest/data-structure/df_state.html) and `df_output` (a pandas DataFrame, full details refer to SuPy documentation at: https://supy.readthedocs.io/en/latest/data-structure/df_output.html), respectively. We note `df_state_final` is designed to have the same data structure as `df_state_init` to allow its reuse as the initial conditions for other SuPy simulations (dashed line, SuPy panel Figure 1).

## 3 Deployment

To achieve cross-platform compatibility, SuPy has two parts:

1) `SuPy_driver` (calculation kernel): the F2PY generated binaries of SUEWS are platform-dependent because of compilation being necessary for assurance of performance.

2) `SuPy` (frontend processor): this platform-independent Python code allows rapid iteration in functionality enhancement and bug-fixing thanks to the powerful ecosystem of Python utilities.

As software compilation can be frustrating and/or prone to operator errors, this procedure is automated using two online services: Microsoft Azure Pipeline (https://azure.microsoft.com/en-us/services/devops/pipelines/) for continuous integration

(CI) and PyPI (https://pypi.org) for distribution. Microsoft Azure Pipeline has good cross-platform support (https://docs.microsoft.com/en-gb/azure/devops/pipelines/agents/) and easy connection with code repositories (e.g., GitHub: https://github.com, Bitbucket: https://bitbucket.com) and supports automated compilation for different platforms.

The Azure Pipeline build workflow permits a variety of functionalities to facilitate compilation and publishing to other online

services (e.g., PyPI, GitHub pages, etc.). Currently, this is setup for three major platforms (Windows, macOS and Linux) with three Python 3 configurations (3.5, 3.6 and 3.7) to conduct automated compilation of SuPy backend files: SUEWS binaries and SuPy_driver, the product of which is directly pushed to PyPI and released in real time.

To build the SuPy_driver two crucial steps to allow cross-platform deployment (full details refer to configuration file

`setup.py` in SuPy_driver) are:

1) *Static linking*: to eliminate the issue of missing dynamic libraries the calculation kernels are pre-built using static linking and therefore run directly after downloading.

2) `manylinux` *tagging*: Given the many Linux distributions and their different runtime libraries that often require distribution-specific compilation, we use the `manylinux` docker image (for details refer to

https://github.com/pypa/manylinux) to compile SuPy_driver.



In addition to the cross-platform compilation, to guarantee delivery quality we perform automatic code tests of four pre-set configurations for every build:

1) *Connectivity between SuPy and SuPy_driver*: checks if the frontend processor and backend calculation core can communicate with correct input and output.

2) *Success in single-timestep mode*: checks SuPy can produce correct simulation results in the single-timestep mode.

3) *Success in multi-timestep mode*: checks SuPy can produce correct simulation results in the multi-timestep mode and does a quick benchmark of computation speed.

4) *Compare simulation results between single- and multi-timesteps modes*: checks SuPy can produce identical simulation results as designed.

All build and test output is logged in detail (see all logs here: https://dev.azure.com/sunt05/SuPy/_build) and the results are reported to developers in real time. This feature is used for all code and underpins a commitment for timely examination of SuPy development.

The Python Package Index (PyPI: https://pypi.org) is the official third-party software repository for Python. As it is supported by the `pip toolchain` it provides Python users easy worldwide access to packages and frees Python developers from maintaining indexing and distribution servers. By using the PyPI channel, SuPy can be easily installed by users with a one-line input in command line tool on any desktop/server system (Listing 1).

**Listing 1: Command line code for SuPy installation using pip. Note Python 3.5+ is required for SuPy installation.**

```
python3 -m pip install supy -U
```

## 4    Demonstration: SuPy Tutorials

To familiarise users with SuPy urban climate modelling and to demonstrate the functionality of SuPy, we provide three tutorials (Table 2, access at: https://supy.readthedocs.io/en/latest/tutorial/tutorial.html) in Jupyter notebooks

(https://jupyter.org/). They can run in browsers (desktop, mobile) either by easy local configuration or on remote servers with pre-set environments (e.g., Google Colaboratory: https://colab.research.google.com, Microsoft Azure Notebooks: https://notebooks.azure.com). In addition, Jupyter notebooks allow great shareability by incorporating source code and detailed notes in one place, which helps users to organise their computation work (Shen, 2014). Jupyter notebooks can be installed with pip on any desktop/server system and open `.ipynb` notebook files locally (Listing 2).






**Listing 2: Command line code for installing Jupyter notebook with pip and open an existing local `.ipynb` notebook file (i.e., `path_to_your_notebook`).**

```
python3 -m pip install jupyter -U
jupyter-notebook path_to_your_notebook
```

**Table 2: Three SuPy tutorials. Note the website links are directed to online Jupyter notebooks for SuPy simulation without any configuration by users.**

| Name | Aim | Audience | Website Link |
| --- | --- | --- | --- |
| Quickstart of SuPy | Essential workflow to conduct SuPy simulations | New users, students | https://mybinder.org/v2/gh/sunt05/SuPy/2019.2?filepath=docs%2Fsource%2Ftutorial%2Fquick-start.ipynb |
| Impact studies using SuPy in Parallel Mode | Impacts on urban climate from varying surface characteristics and forcing conditions | Urban climate researchers with experience in land surface simulations | https://mybinder.org/v2/gh/sunt05/SuPy/2019.2?filepath=docs%2Fsource%2Ftutorial%2Fimpact-studies-parallel.ipynb |
| Interaction between SuPy and external models | Couple SuPy and external models for agile development | Model developers with background in climate modelling | https://mybinder.org/v2/gh/sunt05/SuPy/2019.2?filepath=docs%2Fsource%2Ftutorial%2Fexternal-interaction.ipynb |

To help beginners (e.g., classroom students) run SuPy we provide a forcing dataset (Figure 2) with meteorological data for London in 2012 (Kotthaus and Grimmond, 2014) and an initial model conditions (Table 3) for SuPy. These are made available to SUEWS by calling the `load_SampleData` function. This produces pandas DataFrames with the initial model state (`df_state_init`) and the forcing variables (`df_forcing`). These are used in all the three tutorials.







**Figure 2: Intra-annual (2012) variation of forcing variables in the sample dataset (from top to bottom): incoming solar radiation, air temperature, relative humidity, air pressure, rainfall and wind speed in central London All variables are hourly averages except for total hourly rainfall (source of data: Kotthaus and Grimmond 2014, gap filled: Ward et al. 2016).**



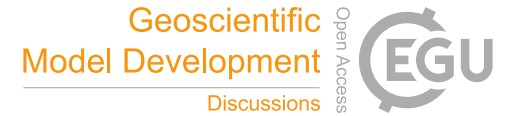

**Table 3: Default settings in the sample dataset provided with SuPy for a) physics scheme and b) basic site characteristics. Full SuPy variable setting details refer to online documentation: https://supy.readthedocs.io/en/latest/data-structure/df_state.html**

| a)   Physics scheme | SuPy variable | Code | Remark |
|---|---|---|---|
| Radiation | `radiationmethod` | 3 | Net all-wave radiation modelled with incoming longwave radiation modelled using air temperature and relative humidity (Loridan et al., 2011) |
| Heat storage | `storageheatmethod` | 1 | OHM model (Grimmond et al., 1991) |
| Anthropogenic heat | `emissionsmethod` | 2 | Anthropogenic heat model responding to temperature, population density, time of day and day of week (Järvi et al., 2011) |
| Snow | `snowuse` | 1 | A snow module for modelling snow pack and related thermal and hydrological dynamics (Järvi et al., 2014) |
| Roughness length for momentum | `roughlenmommethod` | 2 | Momentum roughness length determined using Grimmond and Oke (1999) |
| Roughness length for heat | `roughlenheatmethod` | 2 | Thermal roughness length determined using (Kawai et al., 2009) |
| Atmospheric stability | `stabilitymethod` | 3 | Atmospheric stability correction function (Campbell and Norman, 1998) |

| b)   Basic site characteristics | SuPy variable (unit) | Value |
|---|---|---|
| Land cover fractions for surfaces: building, paved, evergreen tree, deciduous tree, grass, bare soil and water | `Sfr` | [0.43, 0.38, 0.001, 0.019, 0.029, 0.001, 0.14] |
| Building height | `bldgh` (m) | 22.0 |
| Evergreen tree height | `evetreeh` (m) | 13.1 |
| Deciduous tree height | `dectreeh` (m) | 13.1 |

## 4.1   SuPy Quickstart

In this tutorial, we demonstrate the key steps in using SuPy to undertake the core task of SUEWS to simulate energy and water
5   balance in an urban context. Here the runs are for a central London area in 2012.

The surface energy balance (SEB) can be expressed as:

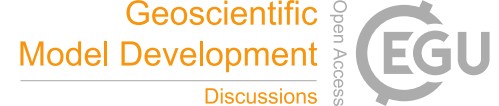

$$Q^* + Q_F = Q_H + Q_E + \Delta Q_S \tag{4}$$

where the flux densities (W m$^{-2}$) are $Q^*$ net all-wave radiation, $Q_F$ anthropogenic heat, $Q_H$ turbulent sensible heat, $Q_E$ latent heat, and $\Delta Q_S$ the net storage heat flux. Through $Q_E$, the SEB characteristics can be linked to the water balance:

$$P + I = E + R + \Delta S \tag{5}$$

where each term is a depth of water per unit of time (e.g. mm d$^{-1}$). $P$ is precipitation, $I$ irrigation, $E$ evapotranspiration (= $Q_E/L_v$ where $L_v$ is the latent heat of vaporisation), $R$ runoff, and $\Delta S$ the net change in water storage.

The fundamental steps to use SuPy after the software environment has been installed (see listing 1 and 2) are: (1) load input, (2) run a simulation and (3) examine the results. With everything ready, three lines of python code are needed (listing 3).

**Listing 3: Python code for a minimal example of SuPy simulation with comments (green).**

```
1    # import supy package
2    Import supy as sp
3    # import sample data
4    df_state_init, df_forcing = sp.load_SampleData()
5    # run supy simulation
6    df_output, df_state_final = sp.run_supy(df_forcing, df_state_init)
```

SuPy is run by calling `run_supy` after `df_state_init` and `df_forcing` have been loaded. After the simulation the two DataFrames provide major SUEWS outputs (`df_output`) and the model state (`df_state_final`) at the end of the run. The latter can be used as initial conditions for other SuPy runs.

15    The post-processing uses `pandas` functions to resample, plot and write out the model output. The default output DataFrame of 5 min resolution can be upscaled to the month for an overview of intra-annual dynamics of surface energy and water balances (Figure 3).





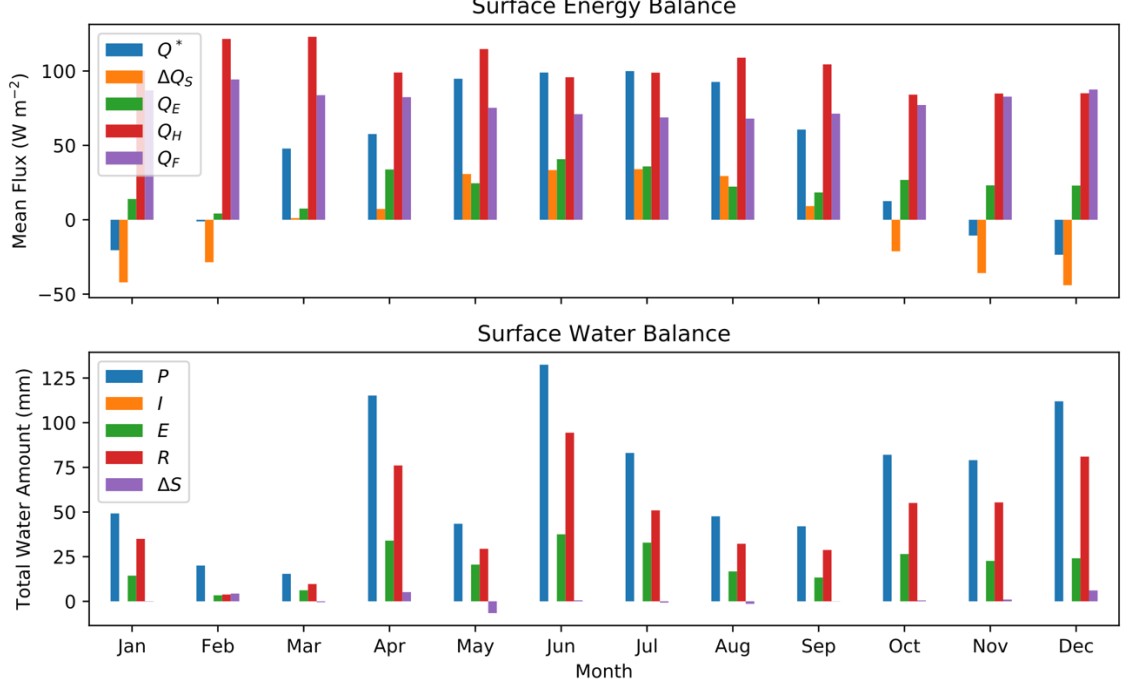

**Figure 3: SuPy simulated monthly (top) surface energy and (bottom) water balance for London 2012. Assuming no irrigation.**

This workflow using SuPy for urban climate modelling can be easily adapted to existing SUEWS tutorials under the UMEP

framework (https://tutorial-docs.readthedocs.io) by replacing the conventional SUEWS binary executable with the python

SuPy package. Given the central role of Python in the UMEP framework, it is expected the adoption of SuPy will further

streamline the workflows for urban climate simulations in UMEP.

## 4.2    Impacts of the urban area on urban climate

A major application of urban climate models is to study the impacts on urban climate from design scenarios that change surface

characteristics or the climate (atmospheric forcing). In this tutorial both scenario types are explored: we provide one example

of modification of albedo for surface characteristics, while another of air temperature alteration for climate conditions.

Technically, this requires several configuration files to be prepared for a suite of independent model runs. These could be run

consecutively (i.e. no interactions between runs are needed) or in parallel, so-called "embarrassingly parallel computation"

(Bailey et al., 1991), with multiple independent runs with sufficient CPUs. In this tutorial, we first demonstrate how SuPy can

be easily setup to efficiently complete multiple simulations in parallel.

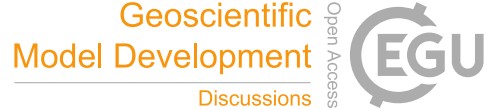

We use dask (https://dask.org) to parallelise the SuPy simulations given its close coherence with `numpy` and `pandas`, in particular its almost identical `DataFrame` interfaces as `pandas`. Specifically, we use the `apply` method of `dask.DataFrame` to improve the simulation performance by distributing the SuPy computations across different configurations. Compared with the serial mode, the `dask`-based parallel mode takes only ~30% of the execution time of the serial mode for simulations longer than 1000 days for 12 grids (Figure 4). The parallel configuration for running SuPy, `run_supy_mgrids`, is then used in the following two cases for more efficient parallel simulations.

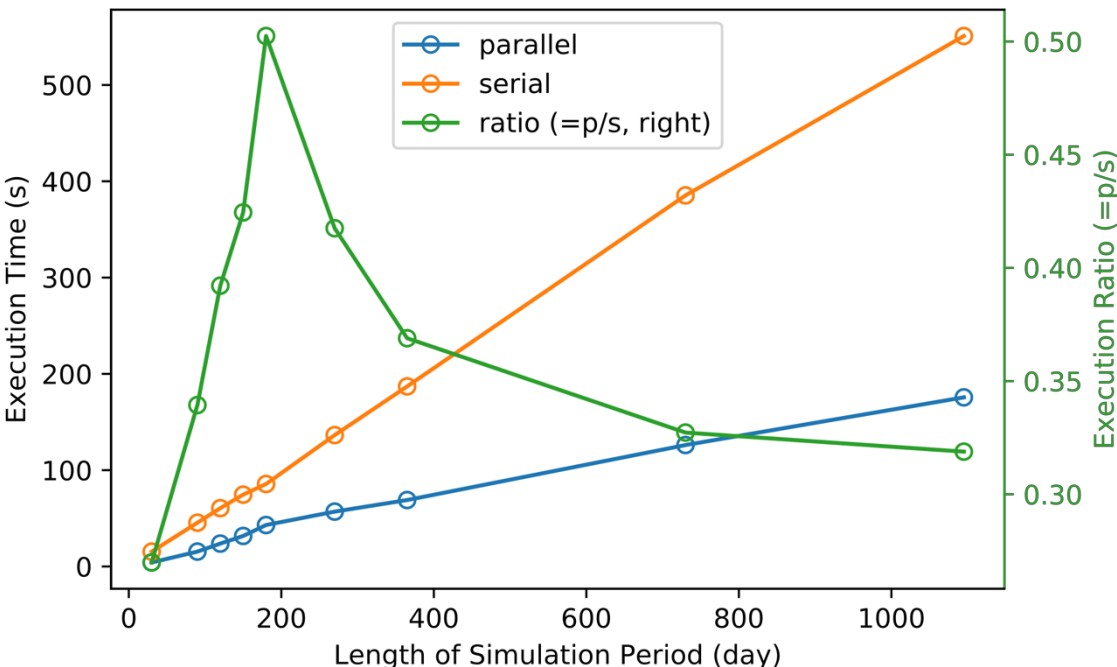

**Figure 4: Comparison of execution times (s) between serial and parallel modes when 12 grids are simulated for different periods (days): 30, 90, 120, 150, 180, 270, 365, 730 and 1095. Simulations performed with macOS 10.14.3 running on 2.9 GHz Intel Core i9 with 32 GB memory. The model configuration is the same as Tutorial 1 (Table 2).**

To explore the effect of changes to surface properties, the DataFrame `df_state_init` needs to be modified. The surface albedo of different materials impacts the outgoing shortwave (solar) radiation and thus the surface energy balance fluxes and other atmospheric variables. Modifying roof albedo has been suggested extensively as a method to cool urban areas (e.g. Li et al., 2014; Ramamurthy et al., 2015; Santamouris et al., 2011). In the example, we conduct simulations from January 2012 to July 2012 with the first 6 months as the spin-up period. The building roof albedo is incrementally increased from 0.1 to 0.8 (e.g. a change of very dark to very light surface). The near surface temperature $T_2$, an indicator of thermal state at pedestrian level, are analysed using the monthly maximum, mean and minimum (Figure 5). It would be expected that the maximum and





mean values of $T_2$ are greatly reduced as they are directly influenced by the altered net solar radiation while impacts on the minimum $T_2$ might be expected to be minimal.

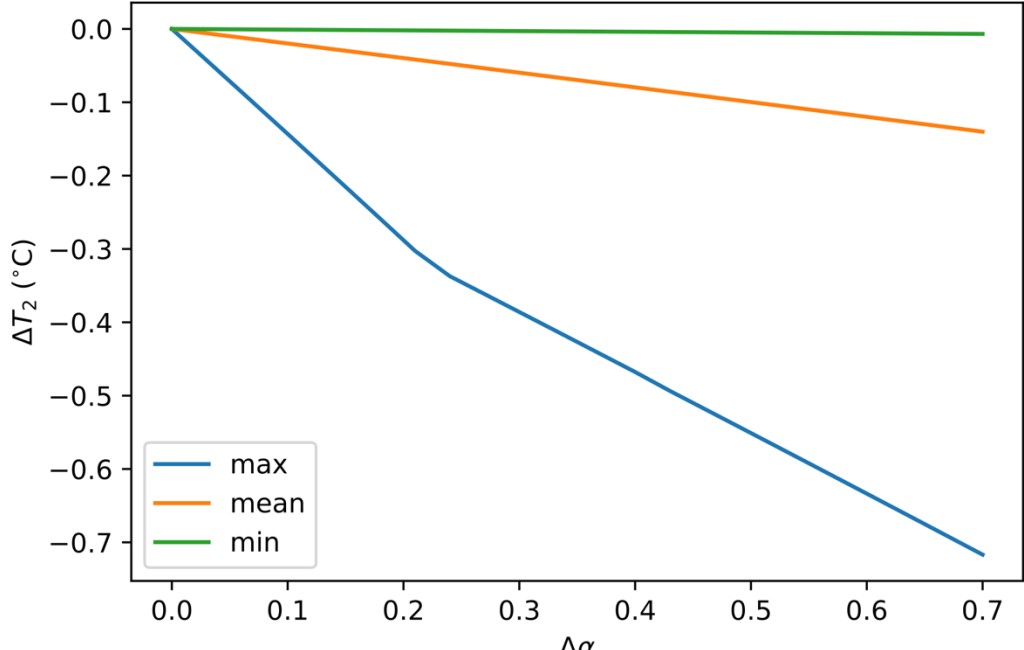

5    **Figure 5: Impacts of increasing building roof albedo $\alpha$ (from 0.1) on near surface temperature $T_2$ considering maximum, mean and minimum daily temperatures at 2 m.**

To explore changes in atmospheric forcing, the DataFrame `df_forcing` is modified. In this example, we investigate the impact of increased local-scale (constant flux layer) air temperature $T_a$ on the near surface air temperature $T_2$. Air temperature

10    $T_a$ is increased over 24 runs from 0 (no change) to +2 °C. The upper limit (+2 °C) represents a highly possible average global warming scenario for the near future (IPCC 2018). The SuPy simulations are conducted January to July 2012 and July data analysed. The $T_2$ results indicate the increased $T_a$ has different impacts on the $T_2$ metrics (minimum, mean and maximum) but all increase linearly with $T_a$. The maximum $T_2$ has the stronger response compared to the other metrics (Figure 6).



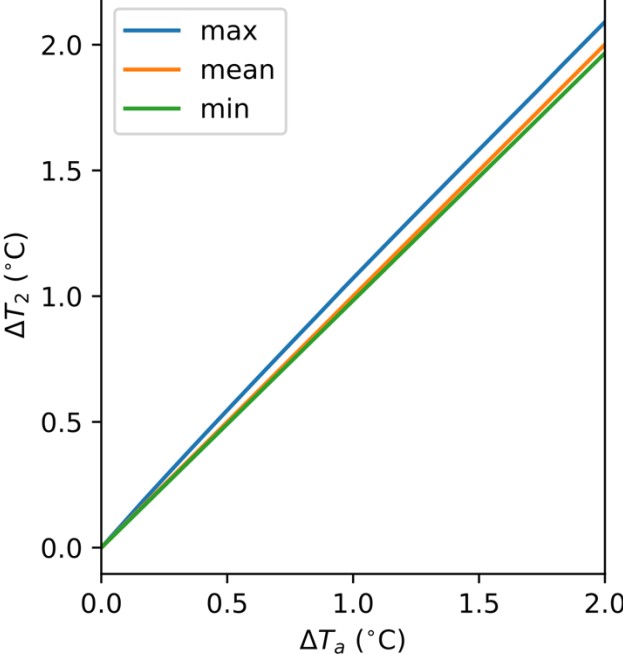

**Figure 6: Impacts of increasing background (constant flux layer) air temperature $T_a$ on near surface maximum, mean and minimum temperatures at 2 m $T_2$. Albedo is 0.1 and land cover characteristics are as Table 2b. Note in this example only one variable is modified.**

This tutorial demonstrates the simplicity of using SuPy to conduct impact studies of both surface characteristics and background climates. These can be easily adapted by users to their specific application interests.

### 4.3    Interaction between SuPy and external models

10   SUEWS can be coupled to other models that provide or require forcing data using the SuPy single timestep running mode (Section 2). We demonstrate this feature with a simple online anthropogenic heat flux model.

Anthropogenic heat flux ($Q_F$) is an additional term to the surface energy balance in urban areas associated with human activities (Gabey et al., 2018; Grimmond, 1992; Nie et al., 2014; 2016; Sailor, 2011). In most cities, the largest emission source is from

15   buildings (Hamilton et al., 2009; Iamarino et al., 2011; Sailor, 2011) and is high dependent on outdoor ambient air temperature. For demonstration purposes we have created a very simple model instead of using the SUEWS $Q_F$ (Järvi et al. 2011) with feedback from outdoor air temperature (Figure 7). The simple $Q_F$ model considers only building heating and cooling:




$$Q_F = \begin{cases} (T_2 - T_C) \times C_B & , T_2 > T_C \\ (T_H - T_2) \times H_B & , T_2 < T_H \\ Q_{F0} \end{cases} \qquad (6)$$

where $T_C$ ($T_H$) is the cooling (heating) threshold temperature of buildings, $C_B$ ($H_B$) is the building cooling (heating) rate, and $Q_{F0}$ is the baseline anthropogenic heat. The parameters used are: $T_C$ ($T_H$) set as 20 °C (10 °C), $C_B$ ($H_B$) set as 1.5 W m$^{-2}$ K$^{-1}$ (3 W m$^{-2}$ K$^{-1}$) and $Q_{F0}$ is set as 0 W m$^{-2}$, implying other building activities (e.g. lightning, water heating, computers) are zero and therefore do not change the temperature or change with temperature.

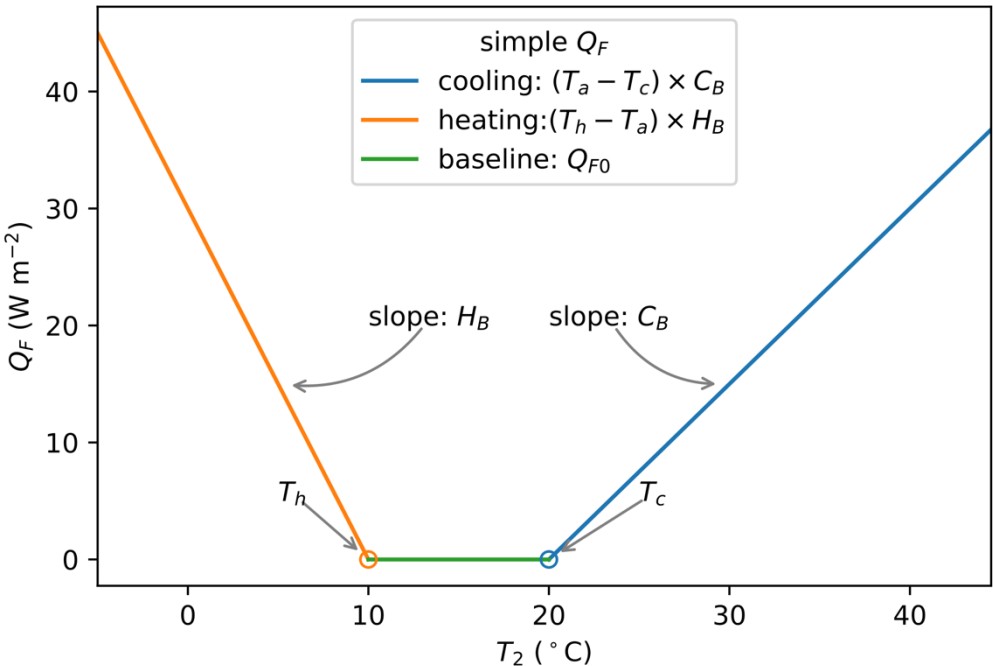

**Figure 7: A simple anthropogenic heat flux ($Q_F$) model as a linear function of air temperature $T_2$.**

The coupling between the simple $Q_F$ model and SuPy is done via the low-level function `suews_cal_tstep`, which is an interface function in charge of communications between SuPy frontend and the calculation kernel. By setting SuPy to receive external $Q_F$ as forcing, at each timestep, the simple $Q_F$ model is driven by the SuPy output $T_2$ and provides SuPy with $Q_F$, which thus forms a two-way coupled loop.

Here we replace the SUEWS $Q_F$ (Table 2) with the simpler $Q_F$ model (Fig. 7, eqn 6) to explore the question of the impact of $Q_F$ on $T_2$ and its feedback on $Q_F$. The simulation using SuPy coupled is performed for London 2012. The data analysed are a summer (July) and a winter (December) month. Initially $Q_F$ is 0 W m$^{-2}$ the $T_2$ is determined and used to determine $Q_{F[1]}$ which in turn modifies $T_{2[1]}$ and therefore modifies $Q_{F[2]}$ and the diagnosed $T_{2[2]}$. Results indicate a positive feedback, as $Q_F$ is





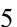

increased there is an elevated $T_2$ but with different magnitudes (Figure 8). Of particular note is the positive feedback loop under warm air temperatures: the anthropogenic heat emissions increase which in turn elevates the outdoor air temperature causing yet more anthropogenic heat release (Figure 8). Note that London is relatively cool (Figure 2) so the enhancement is much less than it would be in warmer cities.

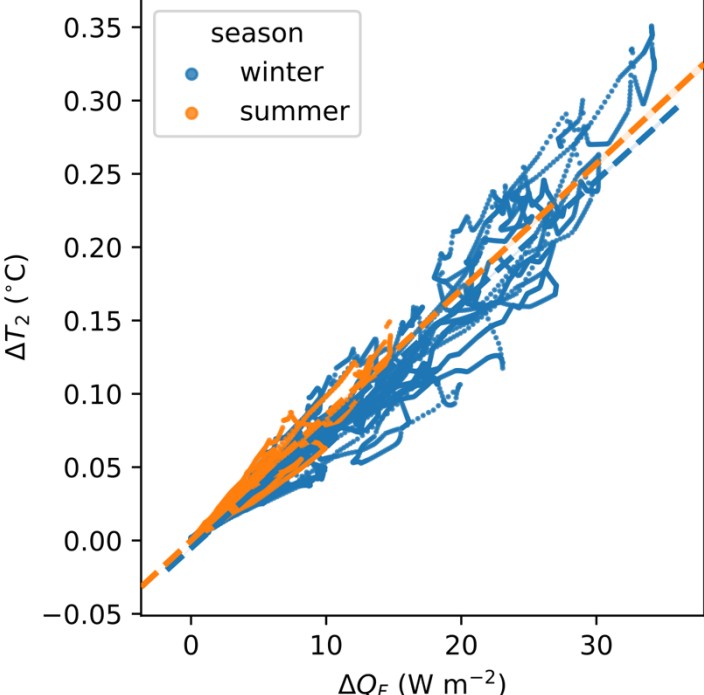

**Figure 8: Impacts of $Q_F$ produced by an external simple anthropogenic heat model on the near surface air temperature $T_2$ for two months: summer (July 2012, orange) and winter (December 2012, blue). Linear regression lines (dashed lines) show the overall seasonal trends. $\Delta Q_F = Q_{F[2]} - Q_{F[1]}$ see text for definitions and the corresponding temperatures $\Delta T_2 = T_{2[2]} - T_{2[1]}$.**

In this case the anthropogenic heat flux model is simple, but a more complex model could be coupled to SUEWS in the same way. This can facilitate development of climate services tools that are both agile and responsive.

## 5    Concluding remarks

15    The development and delivery of a Python-enhanced urban climate model SuPy is introduced, with tutorials (Table 2) to demonstrate typical applications and some new SUEWS features (e.g., surface diagnostics calculation). The Python code and

tutorials are freely and openly available online (see Appendix). Users are encouraged to explore more intriguing urban climate related questions with the enhanced functionalities of SuPy (e.g., flexible configurations, fine control of simulations, etc.).

Notable features of SuPy include:

1) *version consistency via PyPI*: SuPy is distributed via the well managed Python package repository PyPI with all history versions stored. This allows for clear version consistency for reproducing simulation results.

2) *simplicity in input/output sharing*: SuPy uses `pandas DataFrame` as its core data structure and thus draws on a powerful data analysis toolchain, which can facilitate the ease with which urban climate research outcomes can be communicated.

3) *ease of scientific development*: Given the importance of meteorological forcing data in running climate simulations, SuPy

will shortly be equipped with the ability to retrieve forcing variables from global reanalysis datasets. We anticipate data analyses and model development will be added more conveniently within the Python data ecosystem.

4) *an open source tool:* We welcome all kinds of contributions, for example, incorporation of new feature (pull requests), submission of issues, development of new tutorials.

In addition to the SuPy in data analysis and communication features, the computation kernel is SUEWS, so all physics schemes development will remain in the Fortran stack for computational performance and compatibility with a large cohort of scientific code. In one application software, UMEP (Lindberg et al. 2018) written in Python, the SUEWS binary executable will shortly be updated to SuPy for better connectivity to other UMEP components.

We expect SuPy will help guide future development of SUEWS (and similar urban climate models) and enable new applications of the model. For example, the parallel set up of SuPy will allow large scale simulations of urban climate across larger domains with greater surface heterogeneity. Moreover, the improvement in SUEWS model structure and deployment process introduced by the development of SuPy paved the way to a more robust workflow of SUEWS for its sustainable success.

**Appendix A: SuPy model source code and documentation**

Code repository:

- Name: GitHub

- Identifier: https://github.com/sunt05/SuPy

- License: GNU GPL v3.0

- Date first published: 10 February 2019

- DOI: 10.5281/zenodo.2574404





Versioned documentation (tutorials inclusive):

- Name: ReadTheDocs

- Identifier: https://supy.readthedocs.io

- License: GNU GPL v3.0

- Date first published: 10 February 2019

- DOI: 10.5281/zenodo.2599982

## Appendix B: SUEWS software and documentation

Software:

- Name: zenodo

- Identifier: https://zenodo.org/record/2274254

- Available versions: 2018b and newer

- DOI: 10.5281/zenodo.2274254

Versioned documentation:

- Name: ReadTheDocs

- Identifier: https://suews-docs.readthedocs.io

- License: GNU GPL v3.0

- Available versions: 2018b and newer

- DOI: 10.5281/zenodo.2284986

**Code availability:**

Appendix A describes the locations and license information for the SuPy source code and documentation. The source code of calculation kernel SUEWS is available upon request from SG (c.s.grimmond@reading.ac.uk), and its software and

documentation are publicly available at locations provided in Appendix B.

**Author contributions:**

TS led the development of SuPy and core enhancements of SUEWS since v2017b. SG provided overall oversight of the SUEWS development. TS and SG wrote the paper.



**Acknowledgements:**

Support from NERC Independent Research Fellowship (NE/P018637/1; TS); Newton Fund/Met Office CSSP China (SG, TS),

RS Newton Mobility funding (SG, TS), EPSRC LoHCool (SG, TS).

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
