# Peer review of "A Python-enhanced urban land surface model SuPy (SUEWS in Python, v2019.2): development, deployment and demonstration"

_Geoscientific Model Development, 2019_

## Referee Comment (RC1) · Anonymous Referee #1 · 23 Mar 2019

The paper presents a Python-enhanced version of SUEWS model; the original SUEWS model being one of the commonly used urban land surface model in the urban climate research community. The enhanced version SuPy contains quite substantial technical improvement of the original model, including interfacing and pre- and post-processing. The interaction with external models, in particular the incorporation of anthropogenic heat, is also a plus. Overall, the paper is well presented with adequately detailed model structure, documentation, and tutorial demonstration. Minot revisions are needed for some clarification as well as correction of typos and some wordings (listed below). The English writing still need a bit more polish; I leave it for the authors and editor to decide.

[Figure]

Specific comments:

1. p5, line 16, "From similarity theory (Monin and Obukhov, 1954) we estimate", it should be clarified that Eqs. (1)-(3) follows Monin-Obukhov Similarity theory that represents one specific class of similarity theory. The phrase here sounds like MOST is the only similarity theory.

2. p13, section 4.2, "Impacts of the urban area on urban climate", the wording is awkward. Consider revision.

3. p15, Figure 5, why the changes of temperature and albedo follow a linear (bi-linear in the case f maximum temperature) manner?

4. p16, line 15, "high dependent" should be "highly dependent".

5. p17, Figure 7, the notations of threshold temperature T_H and T_C should be made consistent with those in Eq. (6): both in either capital H & C or lower case h & c.

6. p18, Figure 8, the legend "season" is redundant; "summer" and "winter" alone should suffice.

---

## Referee Comment (RC2) · Anonymous Referee #2 · 2 Apr 2019

This paper describes the development of a F2PY version of the land surface model SUEWS and it demonstrates the potential benefits of using F2PY to facilitate usage of the model. It is well written manuscript but some clarifications and adjustments would improve the paper further. I have listed some point below in particular order.

1. It is un-clear what the actual capabilities of SuPy that are available for the user, e.g. is there possibilities for data preparation from surface data in SuPy? I recommend to add a list of the available functions (methods) that is included in SuPy, maybe not in the actual text but as a appendix, depending on the extent of all functions available

2. On page five new model capabilities area presented where pedestrian model output

is added. I recommend that this section is removed as this has nothing to do with the SuPy model per se. If the authors still want this in they should include evaluation of the new features as well, or have a very good reason why this is included here. The three variables (T2, q2 and U10) make use of similarity theory and are estimated within the building canopy layer (close to the ground surface) where this theory should be questioned. Therefore, a detailed evaluation should be added in conjunction with such a model development.

3. There is no reference to the code repository of SuPy in the text. Please add. I found it eventually (https://github.com/sunt05/SuPy) and found that the actual SUEWS source code is not included. Why is this not available as open source from the repository? Most other similar models (WRF, PALM4U, GFDL AM3, SURFEX etc.) have their source code available. Please state why the whole system, both SuPy and SUEWS is not available for other users/developers to be able to contribute to the system. One page 7, line 10, the authors state that using the Python utilities bug-fixing etc. is available. However, since the source code is close I cannot see that this is actually available. Please clarify, or release the full code to the public.

4. Figure 1. Swap around so that it happens in the correct order from a user perspective, i.e. SuPy to the left.

5. Page 8, line 25. What dose 'mobile' mean in this context, cell phone? Why would anyone like to run SuPy on a cell phone? If so, how to use it with other datasets that the one available through SampleData?

6. Page 8, line 25-30. There is a lot of "up-talk" of the system, e.g. "great shearability". This is unnecessary. You are not selling anything. Please go through the text for similar expressions.

7. All tables need formatting. Use right instead of justify alignment.

8. How is SuPy connected to stand-alone Fortran code of SUEWS. Can the same

input data be used in both system. Can SuPy write out data in the same format as the Fortran version of SUEWS? If not, please add this feature so that other systems easily can be used for both Supy and Fortran SUEWS.
* * *

---

## Author Comment (AC1) · 12 Jun 2019

**Response to Reviewer 1**

**General Comments:**

*The paper presents a Python-enhanced version of SUEWS model; the original SUEWS model being one of the commonly used urban land surface model in the urban climate research community. The enhanced version SuPy contains quite substantial technical improvement of the original model, including interfacing and pre- and post-processing. The interaction with external models, in particular the incorporation of anthropogenic heat, is also a plus. Overall, the paper is well presented with adequately detailed model structure, documentation, and tutorial demonstration. Minor revisions are needed for some clarification as well as correction of typos and some wordings (listed below). The English writing still need a bit more polish; I leave it for the authors and editor to decide.*

**Response**: We appreciate the recognition of our work.

**Specific Comments:**
1) *p5, line 16, "From similarity theory (Monin and Obukhov, 1954) we estimate", it should be clarified that Eqs. (1)-(3) follows Monin-Obukhov Similarity theory that represents one specific class of similarity theory. The phrase here sounds like MOST is the only similarity theory.*

**Response**: We agree with the reviewer but as Reviewer 2 suggest we remove this part from this paper, it no longer appears. We will take this suggestion into consideration for future text.

2) *p13, section 4.2, "Impacts of the urban area on urban climate", the wording is awkward. Consider revision.*

**Response**: Changed to "Impacts of urban surfaces on local climate"

3) *p15, Figure 5, why the changes of temperature and albedo follow a linear (bi-linear in the case f maximum temperature) manner?*

**Response**: Note it is not necessarily as the range shows (although the mean suggesting that is the case).

Here the albedo changes K↑ but longwave radiation is not modified because the forcing data are held constant in the simulation. The storage heat flux $\Delta Q_S$ is modified (but not linearly). $Q_F$ is not modified because the forcing data are held constant. The surface conductances will be partially modified (but not fully because the forcing data is being held constant). There will be a nonlinear variation that changes with time as the soil moisture will be modified because of $Q_E$ variations. Thus, there will be a change in $Q_H$ which is used to determine the 2 m air temperature. But the air density and atmospheric stability are using the local scale values in this simulation so are not responding.

Thus the linear relation only holds in this particular case and we would expect this simple simulation to become more non-linear with time. The simulation is holding many things constant at the same time and thus cannot be regarded as being a complete "science" simulation to evaluate this. However, with other feedbacks permitted in the runs their roles could be explored. Some of this become apparent in the later $Q_F$ tutorial.

4) *p16, line 15, "high dependent" should be "highly dependent".*

**Response**: Corrected as suggested.

5) *p17, Figure 7, the notations of threshold temperature T_H and T_C should be made consistent with those in Eq. (6): both in either capital H & C or lower case h & c.*

**Response**: Changed to upper cases for consistency as suggested.

6) *p18, Figure 8, the legend "season" is redundant; "summer" and "winter" alone should suffice.*

**Response**: Removed as suggested.

Responses to Reviewer 2

**General Comments:**

*This paper describes the development of a F2PY version of the land surface model SUEWS and it demonstrates the potential benefits of using F2PY to facilitate usage of the model. It is well written manuscript but some clarifications and adjustments would improve the paper further.*

**Response**: We appreciate the recognition of our work.

**Specific Comments:**

1) *It is un-clear what the actual capabilities of SuPy that are available for the user, e.g. is there possibilities for data preparation from surface data in SuPy? I recommend to add a list of the available functions (methods) that is included in SuPy, maybe not in the actual text but as a appendix, depending on the extent of all functions available*

**Response**: The capacities of SuPy include:

a.  loading existing SUEWS files as SuPy input.
b.  running SUEWS simulations.
c.  saving SuPy results as text files.
d.  visualising SuPy input and output (SUEWS standalone version does not have this capability. Although, the UMEP/SUEWS version does).

A list of `supy` functions is added in Appendix A as suggested.

2) *On page five new model capabilities area presented where pedestrian model output is added. I recommend that this section is removed as this has nothing to do with the SuPy model per se. If the authors still want this in they should include evaluation of the new features as well, or have a very good reason why this is included here. The three variables (T2, q2 and U10) make use of similarity theory and are estimated within the building canopy layer (close to the ground surface) where this theory should be questioned. Therefore, a detailed evaluation should be added in conjunction with such a model development.*

**Response**: Removed as suggested.

3) *There is no reference to the code repository of SuPy in the text. Please add. I found it eventually (https://github.com/sunt05/SuPy) and found that the actual SUEWS source code is not included. Why is this not available as open source from the repository? Most other similar models (WRF, PALM4U, GFDL AM3, SURFEX etc.) have their source code available. Please state why the whole system, both SuPy and SUEWS is not available for other users/developers to be able to contribute to the system. One page 7, line 10, the authors state that using the Python utilities bug-fixing etc. is available. However, since the source code is close I cannot see that this is actually available. Please clarify, or release the full code to the public.*

**Response**:

a.  Inline citations of `supy` code have been added as suggested.

b.  We thank the reviewer for their proposal that contribution to SUEWS/SuPy should be allowed, which is actually what we've been conducting, however, in a different manner from other similar models (e.g., WRF). For SuPy, it is fully open source at GitHub; whereas for SUEWS, we are working on the publication of it as an open source software as well, but at the moment the collaboration needs to be requested upon contact with SG (c.s.grimmond@reading.ac.uk).

4) *Figure 1. Swap around so that it happens in the correct order from a user perspective, i.e. SuPy to the left.*

**Response**: Changed as suggested.

5) *Page 8, line 25. What dose 'mobile' mean in this context, cell phone? Why would anyone like to run SuPy on a cell phone? If so, how to use it with other datasets that the one available through SampleData?*

**Response**: Yes, we meant mobile devices (e.g., cell phones, iPad, etc) that have Internet connections to a remote server running Jupyter services. Also, we would note this feature is NOT implemented by SuPy *per se* but allowed by the Jupyter environment where python 3 is supported. The reason for running SuPy (and many other python applications) on a mobile device (e.g., cell phone) is simple: working seamlessly across different devices is a natural need.

As a discussion on the possibilities allowed by mobile devices is far beyond the scope of this paper, we provide an example: often, we would like to answer *if-so-then-what* questions by changing parameter values inspired by ideas (e.g. classroom, planning). With SuPy, this can be quickly done with a mobile phone (e.g. students in a classroom, practitioner at a site).

To import datasets other than the sample one shipped by SuPy on mobile devices, we would suggest users to try out the following two options:

a. Google Colab: a customised free Jupyter notebook service that has coherent integration with google drive, which thus allows personal files stored on google drive (i.e., SUEWS input files) to be used by SuPy online. Input files other than the sample ones can be uploaded onto Google Drive and thus imported to SuPy under the Google Colab environment.

b. GitHub and other online Jupyter services: setting up a repository with Jupyter Notebooks and data files in junction with online Jupyter services (e.g., binder, CoCalc, etc.) provides a more generic approach to integrate the desktop and mobile workflows conducting SuPy simulations. Notably on iOS, the app Juno provides mobile adaptive interfaces to facilitate the online computing (e.g., SuPy simulations) with Jupyter Notebooks and can incorporate personal files via GitHub repos.

The above comments have been incorporated into the revised manuscript.

6) *Page 8, line 25-30. There is a lot of "up-talk" of the system, e.g. "great shearability". This is unnecessary. You are not selling anything. Please go through the text for similar expressions.*

**Response**: We have removed these.

7) *All tables need formatting. Use right instead of justify alignment.*

**Response**: Changed as suggested.

8) *How is SuPy connected to stand-alone Fortran code of SUEWS. Can the same input data be used in both system. Can SuPy write out data in the same format as the Fortran version of SUEWS? If not, please add this feature so that other systems easily can be used for both Supy and Fortran SUEWS.*

**Response**: Such functionalities have been added in version 2019.5 and a summary of SuPy functions is provided in Appendix A.